In situ SEM/EDS compositional characterization of osteocytes and blood vessels in fossil and extant turtles on untreated bone surfaces; different preservational pathways microns away

Cadena Edwin-Alberto edwin.cadena@urosario.edu.co cadenachelys@gmail.com
Facultad de Ciencias Naturales, Grupo de Investigación Paleontología Neotropical Tradicional y Molecular (PaleoNeo), Universidad del Rosario , Bogotá , Colombia
Smithsonian Tropical Research Institute , Panama City , Panama
Abdala Virginia
Electronic publication date: 2020 Aug 27
Publication date: 2020
Volume: 8
Electronic Location ID: e9833
Received 2020 Jul 3; Accepted 2020 Aug 7
Copyright: ©2020 Cadena
Copyright year: 2020
Copyright holder: Cadena
License: This is an open access article distributed under the terms of the Creative Commons Attribution License, which permits unrestricted use, distribution, reproduction and adaptation in any medium and for any purpose provided that it is properly attributed. For attribution, the original author(s), title, publication source (PeerJ) and either DOI or URL of the article must be cited.
License URL: https://creativecommons.org/licenses/by/4.0/

Keywords: Fossil cells, Exceptional preservation, Osteoblasts, Testudines, Deep time, Mongolia, Colombia, Germany

Funding: Universidad del Rosario, Fondos de Arranque 2018 Code IV-TFA022 Funding for this project was granted to Edwin Alberto Cadena from Universidad del Rosario, Fondos de Arranque 2018 (Code IV-TFA022). The funders had no role in study design, data collection and analysis, decision to publish, or preparation of the manuscript.

==============================
Osteocytes and blood vessels are the main cellular and tissue components of the bone tissue of vertebrates. Evidence of these soft-tissue microstructures has been widely documented in the fossil record of Mesozoic and Cenozoic turtles. However, all these studies have characterized morphologically and elementally these microstructures via isolation from the fossilized bone matrix where they were preserved or in ground sections, which could raise skepticism about the results due to potential cross-contamination or reagents effects. Fossil turtle bones from three different localities with distinct preservation environments and geological settings, including Mongolemys elegans from the Late Cretaceous of Mongolia, Allaeochelys crassesculpta from the Eocene of Germany, and a podocnemidid indet. from the Miocene of Colombia are studied here. Bone from two extant turtle species, Lepidochelys olivacea, and Podocnemis lewyana, as well as a commercial chicken Gallus gallus were used for comparisons. Scanning Electron Microscopy-Energy Dispersive Spectroscopy analyses performed directly on untreated fresh surfaces show that osteocytes-like in the fossil turtle bone are mostly composed of iron and manganese. In contrast, the in situ blood vessels-like of the fossil turtles, as well as those from the extant taxa are rich in elements typically organic in origin (carbon and nitrogen), which are absent to minimally present in the surrounding bone or rock matrix; this suggests a possible endogenous composition for these fossil structures. Also, the results presented here show that although originally both (osteocytes and blood vessels) are organic soft components of bone as evidenced in the extant turtles and chicken, they can experience completely different preservational pathways only microns away from each other in the same fossil bone.

Introduction

Bone is a complex biological tissue that characterizes extant and fossil vertebrates, and consists of a mineralized (calcium, phosphorus) and a non-mineralized (collagen and non-collagenous proteins) extracellular matrix, plus water and some lipids (Boskey & Gehron, 2013; Rey et al., 2009). Cells involved in bone tissue are osteoclasts, osteoblasts, and the most abundant of them osteocytes (Bonewald, 2011). Osteocytes are embedded within the hard-mineralized component of bone throughout life (exceptions being when released by fracture or during remodeling) (Robling & Bonewald, 2020), providing them high preservation potential within fossil bones, which has been extensively documented in different clades of vertebrates (e.g., Bailleul, O’Connor & Schweitzer, 2019; Enlow & Brown, 1956; Pawlicki & Nowogrodzka-Zagorska, 1998; Schweitzer, 2011; Schweitzer et al., 2013; Surmik et al., 2019). Similar preservation of osteocytes- and blood vessels-like has also been documented in fossil turtles, showing that their preservation is independent of geologic time, paleoenvironment, lithology, lineages, and latitude (Cadena, 2016; Cadena, Ksepka & Norell, 2013; Cadena & Schweitzer, 2012; Cadena & Schweitzer, 2014).

Something in common to all aforementioned studies are the analytical tools used to study and characterize these fossil bone microstructures, which include principally: (1) ground sections and observation under transmitted and polarized microscopy (Cadena & Schweitzer, 2012; Surmik et al., 2019); (2) bone demineralization using ethylenediaminetetraacetic acid (EDTA) as a chelating agent (0.5 M, pH 8.0), facilitating release the osteocytes-, blood vessels-, and any other cells- or soft-tissue fibers-like from the bone matrix for their posterior study by transmitted and/or polarized light, scanning and/or transmission electron microscopy and any coupled elemental analyzer, Raman spectroscopy, Fourier-transform infrared spectroscopy (FTIR), immunological and antibody studies (e.g., Alfonso-Rojas & Cadena, 2020; Bailleul, O’Connor & Schweitzer, 2019; Bailleul et al., 2020; Cadena, 2016; Saitta et al., 2019; Schweitzer et al., 2013; Surmik et al., 2019; Wiemann et al., 2018).

The preservation of these soft-tissue microstructures (osteocytes and blood vessels) and their potential original constituents (proteins and DNA) has been questioned and considered a consequence of microbial interactions within fossil bone and its microenvironment or even as a result of cross-contamination in the laboratory (Buckley et al., 2017; Kaye, Gaugler & Sawlowicz, 2008; Saitta et al., 2019). The ‘biofilm hypothesis’ as a source for soft-tissue preservation in dinosaur bones has been rigorously tested, which identified fundamental morphological, chemical and textural differences between the resultant biofilm structures and those derived from dinosaur bone, demonstrating that the recovered microstructures in the reports cited above are endogenous in origin and that the ‘biofilm hypothesis’ should therefore be rejected (Schweitzer, Moyer & Zheng, 2016). Issues concerning cross-contamination and replications, timing of sample collections, and reagents have also been addressed by Schweitzer et al. (2019).

Compositionally, the osteocytes- and blood vessels-like from different clades of fossil vertebrates have been shown to commonly be enriched in iron (Cadena, 2016; Schweitzer et al., 2014; Surmik et al., 2019; Ullmann, Pandya & Nellermoe, 2019), an element that has been suggested to play a key role in preserving and even masking identification of proteins in fossil tissues via Fenton reactions (Schweitzer et al., 2014). Other elements typically found in these fossil bone microstructures are carbon, calcium, and silicon (Cadena, 2016; Ullmann, Pandya & Nellermoe, 2019). At present, all these studies of elemental characterization have been conducted using SEM/EDS on isolated (post-demineralization) osteocytes- and blood vessels-like, or from polished ground sections, which implies some degree of manipulation or contact with reagents or preparation tools, potentially raising skepticism on the elemental results.

Here, I explore the in situ (directly on fresh and untreated surfaces) preservation and elemental composition of bone microstructural elements (cells and blood vessels) of fossil turtle bones from three localities which have completely different geological settings (lithological, taphonomic, and fossil diagenesis), including: (1) Gobi Desert, Mongolia, from the Late Cretaceous (late Campanian–early Maastrichtian); (2) Messel Pit, Germany, from the Eocene; and (3) La Venta fauna, Colombia, from the Miocene. Comparison samples include bone from two extant turtles and a domesticated chicken. I discuss herein the results of these analyses and the advantages of using in situ SEM/EDS for understanding preservation of cells/tissues in fossils.

Materials & Methods

Fossil and extant samples

All the fossil and extant samples analyzed here were free of any resin, glue, or stabilizing additives since field collection. Two small pieces donated by Dr. M Norell (American Museum of Natural History, AMNH) from a partially- articulated shell (carapace and plastron) of Mongolemys elegans (IGM-90/42) were used for this study. Specimen IGM-90/42 has been previously figured, including ground sections that show excellent preservation of osteocytes-like under transmitted light microscopy (Cadena, Ksepka & Norell, 2013, figs. 7, 9). This fossil material was collected by the AMNH and the Mongolian Academy of Sciences joint field expeditions at the Bugin Tsav locality, Gobi Desert, Mongolia, from fine-grained sandstones representing ponds deposits within the Nemegt Formation, considered to be late Campanian–early Maastrichtian (∼80 Ma) in age (Jerzykiewicz, 2000, references therein).

Small isolated fragments from the carapace of an Allaeochelys crassesculpta (SMF ME-2449) were donated by Dr. K Smith (Senckenberg Naturmuseum Frankfurt, SMF); these were collected from the well-known locality of Messel Pit, which represents volcanically-influenced lake deposits from the early-middle Eocene (∼48 Ma) (Lenz et al., 2015). Osteocytes-, blood vessels-, and collagen fibers-like from this specimen were previously described and elementally characterized by Cadena (2016, figs. 4–7).

Carapace fragments from a podocnemidid indet. specimen, (UR-CP-0043), as well as the surrounding rock matrix, were collected in 2018 directly from an excavation site (approximately 1.5 m from the surface) using strict aseptic techniques (nitrile gloves, face mask, wrapped in sterilized aluminum foil and kept in glass containers with silica gel for moisture control until analyses were performed). This fossil material was collected from the Repartidora locality, La Victoria Formation, middle Miocene (13.6 ± 0.2 Ma), Tatacoa Desert, Colombia, from what are interpreted as fluvial deposits (Cadena et al., 2020). Permits for collecting and study of the samples were granted by the Colombian Geological Survey (Radicado No. 20193800017321).

For comparisons, two extant turtle carcasses were sampled directly in the field following the same aseptic protocols used for specimen UR-CP-0043. The first corresponds to carapace fragments from an individual of the sea turtle Lepidochelys olivacea (uncatalogued specimen) collected in January 2017 at the Pacific coast, Santa Elena Province, Ecuador, permit granted by Yachay Tech University. The second (uncatalogued specimen) sampled corresponds to a carcass of the side-necked turtle Podocnemis lewyana found in a sand bed of the Magdalena River, close to La Victoria village, Huila Department, Colombia, under a permit granted by the ethics committee of Universidad del Rosario (Resolución DVO005 672-CV1066) and the Colombian Autoridad Nacional de Licencias Ambientales (Technical concept No. 02263, 2019). A third sample corresponds to a femur fragment from a commercial chicken Gallus gallus obtained directly from a local market. Muscle tissue was removed and small bone fragments were cut using a sterilized scalpel and dried out at room temperature for several days.

Scanning electron microscopy and elemental analysis (SEM/EDS)

Each of the fossils, rock matrix and extant bone samples were placed between two disposable sterilized lab-weighing boats and gently hit with a rock hammer to break them into smaller pieces. Using tweezers (sterilized before every mounting process) one of the smaller pieces of broken bone was transferred to an SEM holder with adjustable screws and secured. To prevent any potential particles or dust from entering the SEM chamber, each sample was gently air cleaned before placing it in the SEM carousel. Elemental analysis was performed in combination with high resolution imaging of the bone surfaces, as well as (in some cases) the rock matrix attached to it using a scanning electron microscope coupled with an energy-dispersive X-ray spectroscopy analyzer (Phenom ProX, at the Paleontological Lab of Yachay Tech University (YTU), San Miguel de Urcuqui, Ecuador). Imaging was performed at 5 kV using different magnification settings, and point-and-map analyses of elemental composition of selected regions or features were performed at 15 kV. At least five or more points were explored for each osteocyte- or blood vessel-like, as well as the surrounding bone matrix or rock. Quality of EDS analyses was evaluated considering only those with one million counts or higher. Full raw data is presented in Data S1.

Bone demineralization

In order to test for the occurrence and preservation of osteocytes- and blood vessels-like in some of the samples, small bone pieces of Mongolemys elegans (IGM-90/42) and the podocnemidid indet. specimen (UR-CP-0043) were demineralized using disodium ethylenediaminetetraacetic acid (EDTA) (0.5 M, pH 8.0 filter-sterilized using a 0.22 μm filter) as previously described (Cadena, 2016; Cleland et al., 2015) for a period of five days to two weeks, or until osteocytes- and blood vessels-like were detected. Photographs of the recovered osteocytes-like were taken using a transmitted light microscope (Olympus BX-63) and a polarized light microscope (Olympus BX-53) at the paleontological lab of YTU. Some of the isolated osteocytes-like from IGM-90/42 were collected with a tip in a 1.5 ml tube, rinsed three times with E-pure water to get rid of EDTA, being centrifuged at 1500 RPM for 2 min between step. A drop of the supernatant was mounted in a stub, dried out at room temperature inside in a sealed small SEM-stub box to avoid any air or dust particles interact with the sample, and analyzed following the same protocol and SEM/EDS machine aforementioned.

Results

Mongolemys elegans Late Cretaceous of Mongolia

The in situ osteocytes-like of Mongolemys elegans (IGM-90/42) under SEM exhibit a distinct contrast with the surrounding bone matrix, which is exclusive of their three-dimensional volume, and it is also different from the empty osteocytes-lacuna, which exhibits the same contrast as the bone matrix (Figs. 1A–1B). Compositionally, they are predominantly composed of iron, calcium, carbon, manganese, and minor amounts of barium and nitrogen (Figs. 1C–1K; 2A; Data S2; and Fig. S1). There is no evidence of any of these elements in empty osteocyte-lacunae walls, which are composed of calcium and phosphorus, like the bone matrix (Figs. 1L–1N). The isolated osteocytes-like show that iron is concentrated on their external surface and the manganese in the internal, this is clearly evident in elemental maps and a cross-line elemental profile (Figs. 1O–1P; Fig. S1). Observation of some of the isolated (post-demineralization) osteocytes-like under transmitted and polarized light revealed excellent morphological preservation, with some of them emitting low-degree birefringence colors under polarized light (Fig. 3).

Allaeochelys crassesculpta, Eocene of Germany

The most abundant bone microstructures preserved in this sample are blood vessels-like and the walls that formed the Haversian-Volkmann-like (H-V) canals; also, in some, there is evidence of very small (2.5 µm diameter) structures with a striated margin which resemble the morphology of osteoblast cells (Figs. 4A–4D; Fig. S2). The blood vessels-like exhibit a width of 1–3 µm, with an average wall thickness of 0.2 µm (Fig. 4D). Compositionally, the blood vessels-like are mainly composed of carbon and nitrogen, with minor amounts of calcium, phosphorus and iron (Fig. 2B; 4E–4G; 4J –4N; Fig. S2; Data S2). The bone matrix surrounding them lacks nitrogen and carbon, and it is exclusively characterized by calcium, phosphorus, and iron (Figs. 4E, 4I). A bone sample with rock matrix attached shows that the bone is composed of calcium, phosphorus, iron, and nitrogen, and, in contrast, the rock matrix is rich in aluminum and silicon (Figs. 2C–2D; Figs. 4O–4Q; Data S2).

Figure 1 SEM/EDS analyses of Mongolemys elegans (IGM-90/42) bone.

(A) Micrograph of one of the osteocytes-like and an empty lacuna nearby. (B) EDS of the bone region shown in (A), in which orange indicates bone matrix (calcium), and blue-yellow denotes the osteocyte-like (oxygen and iron respectively). (C) Micrograph of one osteocyte-like, indicating the regions where EDS mapping and point analyses were performed. (D–E) Composite elemental map (D) and individual element maps (E) for the rectangle labeled as 1 in (C), in which osteocytes-like show high amount of iron and nitrogen. (F) Elemental point values of point 7 (bone matrix) shown in (C). (G) Elemental intensities for point 4 (osteocyte-like) shown in (C). (H) Micrograph of a broken osteocyte-like inside its lacuna. (I) Individual elements maps from rectangle 1 shown in (H), with the broken osteocyte-like showing a high content of manganese. (J) Elemental intensities for point 2 (bone matrix) shown in (H). (K) Elemental intensities for point 6 (osteocyte-like) shown in (H). (L) Micrograph of a empty lacuna. (M–N) Composite elemental map and individual elements maps for the rectangle labeled as 1 in (L), in which the wall surface of the lacuna exhibits the same composition as the bone matrix. (O) An isolated, broken, and folded osteocyte-like showing a high amount of iron at its external surface and manganese in its internal region, the red line denotes the cross-line described in (P). (P) Cross-line elemental profile across the broken and folded osteocyte-like shown in (O), revealing a switch between iron and manganese content between its external and internal surfaces. Full EDS results for the points shown in (C, H, and L) are presented in Fig. 2 and Data S2.

Podocnemidid indet, Miocene of Colombia

The sample of the side-necked turtle from La Venta, Colombia, shows on the bone external cortex preservation of walls that formed the H-V-canals, blood vessels- and osteocytes-like tightly embedded in the very homogenous bone matrix (Figs. 5A–5B; 5F–5G). Elementally, the blood vessels-like and H-V-canal walls are rich in carbon, nitrogen, and calcium, with minor amounts of phosphorous and silicon (Figs. 2B; 5C–5D; 5H–5I; Data S2). In contrast, the osteocytes-like are composed of iron, calcium, aluminum, manganese, phosphorus, and minor amounts of silicon (Figs. 2B; 5H–5J; Data S2). The bone matrix lacks carbon and nitrogen, and it is constituted by calcium and phosphorus mainly (Figs. 2C; 5C, 5E 5H). An isolated bone fragment (post-demineralization) shows some of the osteocytes-like still embedded in the matrix, varying in color from orange to black, the darker ones located closer to black, dendritic mats (Figs. 5K–5M).

Figure 2 %Wt of elements for fossil and extant turtle bone.

(A) EDS point analyses of osteocytes, indicating high amounts of iron and manganese in the fossil cells (from Mongolemys elegans and the podocnemidid indet. specimens), whereas carbon and nitrogen dominate the cells from extant taxa (Podocnemis lewyana and Lepidochelys olivacea), including those from a chicken (Gallus gallus). (B) EDS point analyses of blood vessels, with fossil and extant showing a similar elemental composition rich in carbon and nitrogen. (C) EDS point analyses of the bone matrix surrounding osteocytes or blood vessels. Note how fossil and extant samples exhibit similar %Wt values for calcium, carbon, and phosphorus. Fossils show relative enrichment in iron and a lower amount of nitrogen in comparison to the extant bone matrix samples. (D) EDS point analyses of the surrounding rock matrix, which show an absence of carbon, calcium, and nitrogen, but abundant silicon and aluminum. Full data for these EDS point analyses are presented in Data S1, S2.

Figure 3 Fossil osteocytes-like from Mongolemys elegans (IGM-90/42).

(A–F) Isolated (post-demineralization) osteocytes-like viewed under transmitted light microscopy (A, C, E) and polarized light microscopy (B, D, F), showing low to moderate birefringence. All photographs taken with a 100X-oil immersion objective lens.

Figure 4 SEM/EDS analyses of Allaeochelys crassesculpta (SMF ME-2449) bone.

(A–D) Micrographs of two Haversian canals, in which (B, D) show the blood vessels-like outlined in red, and osteoblasts-like outlined in green in. Measurements of the width of a blood vessel-like width, wall thickness, and osteoblast-like diameters are shown in (D). (E–G) Micrograph (E) and EDS elemental maps (F–G) of one of the blood vessels-like. (H) Elemental intensities for point 3 (blood vessel-like) shown in (E), showing it to be rich in carbon and nitrogen. (I) Elemental intensities for point 5 (bone matrix) shown in (E), showing an abundance of calcium, phosphorus, and iron, and absence of carbon and nitrogen. (J) Bone fragment placed in the SEM holder. (K) Micrograph showing a blood vessel-like embedded in the bone matrix, from the yellow region indicated in (J). (L) Close-up micrograph of the blood vessel-like shown in the red rectangle in (K). (M) Elemental intensities for point 3 (blood vessel-like) shown in (L), showing a high amount of carbon. (N) Elemental maps of the blood vessel-like shown in (L), indicating a high amount of carbon and nitrogen, these elements are absent in the surrounding bone matrix, which is composed mainly of calcium and phosphorus. (O–Q) Micrograph and EDS elemental maps of a bone margin in contact with rock matrix, the latter of which exhibits relatively higher amount of aluminum and absence of calcium, phosphorus, and iron. Full data for these EDS point analyses are presented in Data S1, S2.

Figure 5 SEM/EDS analyses of podocnemidid indet. (UR-CP-0043) bone.

(A) Bone sample mounted in the SEM holder. (B–C) Micrograph and EDS element maps of one of the blood vessels-like embedded in the bone matrix, showing high amounts of carbon and nitrogen, a moderate amount of silicon; this differs from the bone matrix, which is dominated by calcium and phosphorus. (D) Elemental intensities for point 2 (blood vessel-like) shown in (B). (E) Elemental intensities for point 5 (bone matrix) shown in (B), showing a high amount of calcium and phosphorus. (F–G) Micrographs showing an osteocyte- and blood vessel-like 20 μm away from each other, both embedded in the bone matrix. (H) EDS element maps of the region shown in (F), the blood vessel-like exhibits high amounts of carbon and nitrogen, and the osteocyte-like is richer in iron but lacks significant carbon and nitrogen. (I) Elemental intensities for point 2 (blood vessel-like) shown in (F). (J) Elemental intensities for point 7 (osteocyte-like) shown in (F). (K) An isolated bone fragment after four days of demineralization, viewed under transmitted light microscopy (at 20×). (L) Close-up of the red rectangle region shown in (K), under polarized light microscopy (at 40×), showing darker osteocyte-like adjacent to where dendritic pyrolusite mats. (M) Close-up of the red rectangle region shown in (L) (at 100×), showing the dendritic pyrolusite and some of the osteocytes-like in detail. Full data for these EDS point analyses are presented in Data S1, S2.

In situ extant turtle and chicken bone microstructures

The carapace bone fragment of the extant side-necked turtle Podocnemis lewyana shows osteocytes within lacunae (Figs. 6A–6B). Their composition is rich in carbon, nitrogen, calcium, and phosphorus (Fig. 2A; 6C–6D; Data S2). The bone matrix is relatively richer in calcium (Fig. 2C; 6C, 6E). The H-V canals exhibit a distinct wall and a high concentration of blood vessels and red blood cells, which are rich in carbon and nitrogen (Fig. 2B; 6F–6G; Figs. S3; S4; Data S2). Similar spatial patterns and composition are shared by the bone of the extant marine turtle Lepidochelys olivacea (Fig. 2; 6H–6M; Data S2), and the bone of Gallus gallus (chicken) (Fig. 2; 6N–6P; Data S2).

Figure 6 SEM/EDS analyses of the extant turtle and chicken bones.

(A) Bone fragment of Podocnemis lewyana in the SEM holder. (B–C) Micrograph and EDS element maps of osteocytes embedded in the bone matrix, showing high amounts of carbon and nitrogen; this differ from the bone matrix, which is dominated by calcium and phosphorus. (D) Elemental intensities for point 3 (osteocyte) shown in (B). (E) Elemental intensities for point 4 (bone matrix) shown in (B). (F) Micrograph of one of the Volkmann canals and a blood vessel system in the sample of P. lewyana. (G) Close-up of the Volkmann canal wall and blood vessel system (outlined in red) P. lewyana. (H) Bone fragment of Lepidochelys olivacea in the SEM holder. (I) Micrograph of a region of the cancellous bone shown in the yellow rectangle in (H). (J–K) EDS composite (J) and individual elemental (K) analyses of the bone region shown in (I). (L) Elemental intensities for point 2 (blood vessel) shown in (I), showing high amount of carbon and nitrogen. (M) Elemental point values for point 5 (bone matrix) shown in (I), showing high amounts of calcium and carbon. (N) Bone fragment from a femur of Gallus gallus in the SEM holder. (O) Micrograph of the bone region shown in the yellow rectangle in (N), showing osteocytes embedded in the bone matrix. (P) EDS elemental maps of one of the osteocytes (in the red rectangle) shown in (O), showing a high amount of carbon within the cell. Full data for these point analyses are presented in Data S1, S2.

Discussion

As previously shown (Cadena, 2016; Schweitzer et al., 2014; Surmik et al., 2019; Ullmann, Pandya & Nellermoe, 2019), the in situ analyses presented here, concur with that iron is a very common constituent of fossil osteocytes-like, such as those found in the Late Cretaceous Mongolemys elegans and the Miocene podocnemidid indet. bone samples studied herein (Figs. 1 and 5). However, this composition is not always homogenous and may vary between the external and the internal layer of osteocytes-like, as shown in a broken and folded osteocyte-like from M. elegans, which exhibits richer content of manganese internally and iron externally (Figs. 1O–1P). High levels of manganese were also detected in osteocytes-like from the Miocene side-necked turtle from Colombia, indicating that besides iron as initially suggested by Schweitzer et al. (2014), manganese may also be involved in the preservation of these bone microstructures in deep time. The source for this rich content of manganese seems to be from manganese oxides such as pyrolusite penetrating bone microfractures, which were found herein in some fragments of the Miocene podocnemidid indet. from Colombia (Figs. 5K–5M), and also has been characterized to occur in dinosaur fossil bones from the same Nemegt Formation, from which the M. elegans studied herein was collected (Owocki et al., 2016). The color variation exhibited by the fossil osteocytes-like of M. elegans and podocnemidid indet. seems to be related to enrichment of manganese, higher their manganese content darker their color. In contrast to the osteocytes of the extant turtle and chicken bone, which are rich in carbon and nitrogen (Figs. 2 and 6), these elements only appear in minor amounts in fossil osteocytes-like. However, these cells exhibited a very distinct composition when compared to the surrounding bone matrix and even the wall surfaces of their osteocytes-lacunae, indicating that their mineralized preservation occurred at micro-scale inside the bone, a hypothesis that should be tested by future studies using additional tools (e.g., Raman and FTIR spectroscopy).

The blood vessels- and H-V canal walls-like preserved in the Eocene Allaeochelys crassesculpta from the Messel Pit and the Miocene podocnemidid indet. specimen from the La Venta not only exhibited a similar morphology, but also exhibited the same elemental composition as their corresponding tissues in extant turtle and chicken bone. (Figs. 2, 4 and 5, Figs. S2– Figs. S4; Data S2). In both cases (extant and fossils) being rich in carbon and nitrogen, and differing from the surrounding bone matrix which is richer in calcium and phosphorus, or the rock matrix which is rich in silicon and aluminum (without any traces of carbon, calcium, or nitrogen) which suggests that carbonates or nitrates were absent in the surrounding microenvironment. The in situ measurements performed on some of the preserved blood vessels-like from A. crassesculpta, exhibiting uniform fabric and thin walls of 0.2 µm thickness (Fig. 4D) suggest that they are not consistent with the characteristics of biofilms, which tend to be amorphous and larger in diameter (Schweitzer, Moyer & Zheng, 2016). Blood vessels constitute one of the most promising microstructures preserved in fossil turtles for molecular paleontology studies, and future studies should focus on their molecular in situ characterization using ToF-SIMS mass spectrometry, similarly as it has been used in dinosaurs and other fossil vertebrates (Alfonso-Rojas & Cadena, 2020; Henss et al., 2013; Lindgren et al., 2018; Schweitzer et al., 2019).

For the first time, I herein report the preservation of osteoblasts-like in fossil vertebrates, particularly in the Messel Pit turtle A. crassesculpta. They occur as oval objects with striated margins that are attached to the H-V canals (Figs. 5B–5D; Fig. S2), and thus resemble the morphology and size of osteblasts observed in electron micrographs of human bone (Nakamura, 2007; Schmidt et al., 2002). At the same time, evidence here provided from the Miocene podocnemidid indet. turtle from Colombia (Figs. 5F–5J) shows that, in the same bone specimen, osteocytes- and blood vessels-like that are only 20 µm away from each other are compositionally different. This indicates that each microstructure went through a different preservational pathway. Osteocytes-like seem to be more mineralized than blood vessels-like in these fossil samples, with high amount of iron and manganese, and less organic components than blood vessels-like (Figs. 2 and 3). In the extant bone of turtles and chicken, osteocytes and blood vessels exhibit similar elemental composition under SEM/EDS, both being rich in carbon and nitrogen, which are typically present in abundance within proteins (Torabizadeh, 2011) (Figs. 2 and 6). A similar composition was detected herein in fossil blood vessels from A. crassesculpta from Germany and the podocnemidid indet., from Colombia (Figs. 4 and 5).

Traditionally, it has been suggested that SEM/EDS has to be performed on homogenous or polished surfaces to avoid topographic effects on EDS analyses (Goldstein et al., 2003). However, as I showed here, such effects were negligible for the analyzed samples with composition and signal intensities being very similar in both the fossil and extant samples (Fig. 2). I therefore suggest that a more critical condition for EDS analysis on untreated samples is acquire the highest maximum count rate possible; above 1 million counts is ideal.

Conclusion

This study provided evidence that in situ analyses using a conventional technique, SEM/EDS, on untreated fresh surfaces of fossil and extant bones constitutes a protocol that should be added to the rigorous plethora of proxies and tools (e.g., those recently reviewed and summarized by Schweitzer et al. (2019) to support and demonstrate the preservation of cells, soft-tissues and their original constituents in deep time. Furthermore, in situ analyses of fossil and extant bone samples may also help eliminate any potential skepticism of results obtained by molecular paleontology studies, because, as demonstrate here, it requires minimal sample preparation/manipulation, use of reagents, or contact with lab tools that could cause possible contamination.

Supplemental Information

Supplemental Information 1 Raw data from SEM/EDS including figures, spectrum, EDS analyses for all fossil and extant samples studied

Click here for additional data file.

Supplemental Information 2 Summary of the EDS elemental point analyses (%Wt) shown in Fig. 2

Click here for additional data file.

Supplemental Information 3 Figures S1-S4

Additional figures of fossil and extant samples studied here.

Click here for additional data file.

I thank M Norell, K Smith, S Schaal and A Vanegas for access to samples. I also thank M Schweitzer for some preliminary feedback on some of the results presented here. Thanks are also extended to Yachay Tech and the Colombian Geological Survey and Ethics Committee at Universidad del Rosario for the permits to collect and analyze the samples. Thanks to P Ullmann and another anonymous reviewer for comments that improved this manuscript.

Institutional abbreviations

AMNH American Museum of Natural History, New York, USA

IGM Geological Institute of the Mongolian Academy of Sciences, Ulaan Baatar, Mongolia

SMF ME Senckenberg Naturmuseum Frankfurt, Germany

UR-CP paleontological collection, Facultad de Ciencias Naturales, Universidad del Rosario, Bogotá, Colombia

Additional Information and Declarations

Competing Interests

Author Contributions

Ethics

Field Study Permissions

Data Availability

The author declares that he has no competing interests.

Edwin-Alberto Cadena conceived and designed the experiments, performed the experiments, analyzed the data, prepared figures and/or tables, authored or reviewed drafts of the paper, and approved the final draft.

The following information was supplied relating to ethical approvals (i.e., approving body and any reference numbers):

Ethics committee of Universidad del Rosario granted approval to the project and experiments conducted (DVO005 672-CV1066).

The following information was supplied relating to field study approvals (i.e., approving body and any reference numbers):

Sample collecting and analyses were approved by the following institutions: (1) Yachay Tech, University, Ecuador. (2) Autoridad Nacional de Licencias Ambientales (Technical concept N0 02263, 2019), Bogotá, Colombia. (3) Colombian Geological Survey (Radicado N0 20193800017321), Bogotá, Colombia. (4) American Museum of Natural History, New York, USA, and (5) Senckenberg Naturmuseum Frankfurt, Germany.

The following information was supplied regarding data availability:

The raw data is available as a Supplementary Files.

All the fragment specimens studied are stored at the UR-CP, paleontological collection, Facultad de Ciencias Naturales, Universidad del Rosario, Bogotá, Colombia. For access, they all are stored in a single box identified under the label UR-CP-SEM-in situ-turtles.

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
