# Peer review of "In situ SEM/EDS compositional characterization of osteocytes and blood vessels in fossil and extant turtles on untreated bone surfaces; different preservational pathways microns away"

_PeerJ, doi:10.7717/peerj.9833_

## Round 0.1 · original submission · Minor Revisions

The two reviews received suggest minor revisions. I would like you to name osteocytes, osteoblasts, and blood vessels with quotation marks around them or add a ‘-like’ term after the word, to highlight that more advanced biochemical procedures should be used to confirm their identities. It is also needed to make the statements suggested by our first reviewer less intense. In this first review an annotated pdf was attached, please pay attention to all the suggestion included in it.
Please clarify the point of the permits for the use of the specimens.

·

Basic reporting

Some sentences could use rephrasing to improve their ease of readability and to provide better clarity on some of the methods, results, or intended meaning. I’ve noted where changes would be most helpful with comments, and have provided small in-text suggestions that could augment the text’s readability.

‘Groups’ should instead be referred to as ‘clades’.

Do the grouped citations each need to be rearranged into chronological order?

The author might consider adding a couple citations to a published doctoral thesis (Boles 2016) that also explores preservation and composition of cells and tissues within fossil turtle bones via demineralization and EDX.

Many of the figure captions could use clarification/further details concerning labeling, explanation of color uses and symbols in the figures, and further explanation of how to interpret them. I may know how to read/interpret most of them visually, but not every reader will, so adding a few details would be beneficial. I’ve attempted to identify questions that can be answered through such (short) additions, and to provide suggested edits to improve reader comprehension of them.

Experimental design

A bit more detail on the treatment, transfer, and imaging protocols used with demineralization-isolated cells would be helpful to less-informed readers looking to try this type of study themselves. See my Comment #27 in the attached PDF for details.

Validity of the findings

Since no biochemical methods have been used to rigorously test the endogeneity of the recovered microstructures and identification was guided instead by morphology and gross elemental composition, osteocytes, osteoblasts, and blood vessels should all technically be referred to with either single or double quotation marks around them, or with the addition of a ‘-like’ term after the word (e.g., ‘osteocytes’, or osteocyte-like microstructures). This is standard practice for this topic, and is a requirement until they are shown via more advanced biochemical methods to truly be endogenous in molecular structure/composition.

In a couple spots in the Discussion the statements seem to be written in a bit too strong of language. It’s not over-interpretation of results, but rather how the verbs are used in a few sentences. It would be appropriate to simply dial back the statements applying toward other microstructures recovered by previous and/or future studies, such as using “may vary” instead of saying it simply “varies” (line 256), or using “may also be involved” rather than “is involved” (line 261). Those statements might be true with the samples studied here, but might not be universally true concerning similar microstructures identified in other previous and future reports. So, I think it would be better to use more conservative language. My Comments #41, 51, and 56 in the attached review PDF note similar instances.

Reviewer 2 ·

Basic reporting

No comment.

Experimental design

For all specimens, especially the Mongolian ones, were all permits obtained?

Validity of the findings

No comment.

Additional comments

In this manuscript, Cadena examines non-demineralized turtle bones from a range of ages and regions around the world to examine elemental composition of soft tissue preservation. This study fills a gap between elemental characterization of soft tissues from demineralized fossil material and clearly shows the restricted nature of the elemental composition leading to preservation.

Please make sure that all figures are colorblind accessible. For example, Figure S2 has multiple areas with both red and green figures. If there are questions of accessibility, https://www.color-blindness.com/coblis-color-blindness-simulator/ works quite well.

---

## Round 0.2 · accepted · Accept

Thank you for having all the reviewer suggestions into consideration. We are ready to move on.